# Gender Difference in Orthostatic Vascular Stiffness Increase in Young Subjects

**DOI:** 10.3390/diagnostics15050517

**Published:** 2025-02-20

**Authors:** Victor N. Dorogovtsev, Dmitry S. Yankevich, Valentina M. Tsareva, Denis A. Punin, Ilya V. Borisov, Natalya N. Dekhnich, Andrey V. Grechko

**Affiliations:** 1Federal Research and Clinical Center of Intensive Care Medicine and Rehabilitology, 107031 Moscow, Russia; dyankevich@fnkcrr.ru (D.S.Y.); realzel@gmail.com (I.V.B.); noo@fnkcrr.ru (A.V.G.); 2Smolensk State Medical University, 214019 Smolensk, Russia; tsarev.al@mail.ru (V.M.T.); pun.92.work@gmail.com (D.A.P.); n.dekhnich@mail.ru (N.N.D.)

**Keywords:** head-up tilt test, pulse wave velocity, functional reserve, orthostatic regulation of circulation, orthostatic changes in baPWV, gender differences, vascular stiffness, vascular ageing, arterial hypertension

## Abstract

**Background/Objectives:** Early detection of increased vascular stiffness in young populations may facilitate the development of more effective strategies for the primary prevention of arterial hypertension and other age-related cardiovascular diseases. To examine gender differences in orthostatic increases in vascular stiffness during the head-up tilt test (HUTT), standardized by hydrostatic column height. **Materials and Methods:** A total of 133 healthy adults aged 18–20 years (93 females and 40 males) were evaluated. Blood pressure and pulse wave velocity at the brachial–ankle artery site (baPWV) were measured using an ABI system 100 PWV multichannel sphygmomanometer. Orthostatic changes in arterial stiffness were assessed during a head-up tilt test (HUTT) using the Luanda protocol, which standardizes hydrostatic column height. The functional reserve coefficient (FRC) of orthostatic circulatory regulation was introduced as a measure of adaptive capacity: FRC = ΔbaPWV/baPWVb. This coefficient accounts for both structural (baPWVb) and functional (ΔbaPWV = baPWVt − baPWVb) components influencing cardiovascular system adaptation, which exhibit multidirectional changes with age. **Results:** Baseline baPWV (baPWVb) values in the horizontal position showed no significant differences between genders and were within normal age ranges. However, baPWV values in the upright HUTT position (baPWVt) were significantly higher in men (*p* = 0.0007). Dynamic biomarkers of vascular reserve, including ΔbaPWV and FRC, were also significantly elevated in men (*p* = 0.0009 and *p* = 0.0064, respectively). **Conclusions:** While baseline baPWVb values were comparable between genders, dynamic biomarkers of vascular reserve, such as ΔbaPWV and FRC, were significantly higher in men. Prospective studies are needed to establish optimal reference values for these dynamic biomarkers, enabling the assessment of individual trends in vascular aging and evaluating the effects of treatment, lifestyle modifications, and other preventive measures on vascular health.

## 1. Introduction

Increased vascular stiffness is one of the most important risk factors for the development of arterial hypertension (AH) and other age-related cardiovascular diseases (CVD), as well as chronic kidney disease (CKD) [1,2,3,4,5]. To improve the effectiveness of primary prevention of CVD, it is necessary to develop approaches that allow the diagnosis of processes that precede structural changes in the arteries and an increase in their arterial stiffness at an early preclinical stage.

Clinical orthostatic disturbances: Clinical orthostatic hypotension (OH) and clinical orthostatic hypertension (OHT) are also risk factors for the development of AH and other age-related CVD [6,7,8,9]. Clinical OH has been shown to be associated with an earlier increase in vascular stiffness compared with orthostatic normotension (ONT) [10,11,12]. Preclinical OH and OHT have been considered a variant of normal, but it has been shown that in the older age group, individuals with asymptomatic preclinical OH have accelerated vascular wall remodeling with thickening of IMT [13]. In a young population under 30 years of age, subjects with preclinical OH significantly increase the risk of developing AH, but this is also true in the older population [14].

To study asymptomatic preclinical orthostatic abnormalities, we developed a specific passive orthostatic testing (HUTT) protocol (Luanda protocol) with standardization of the hydrostatic column height [15]. This was necessary to equalize differences in the subject’s height. In our recent study, we demonstrated age-related changes in arterial stiffening measured by pulse wave velocity at the brachial and ankle artery site (baPWV) in three age groups of conventionally healthy subjects: under 30 years, 30 to 45 years, and over 45 years [16]. We demonstrated an age-related increase in baPWV, which is consistent with the literature data [17]. To obtain additional information reflecting the state of orthostatic regulation of hemodynamics, we proposed new indices: functional reserve of orthostatic regulation of hemodynamics ΔbaPWV—increase in the index during HUTT: ΔbaPWV = baPWVt − baPWVb and functional reserve coefficient FRC = ΔbaPWV/baPWVb. The age dynamics of these indices are characterized by a progressive increase in the level of baPWVb and a decrease in functional reserve (ΔbaPWV) and FRC [18]. This study allows us to identify the main trends in vascular aging: progressive increase in baseline PWV and progressive decrease in functional reserve. Of particular interest is the study of gender differences in vascular aging parameters at an early preclinical stage in young healthy subjects before the development of signs of vascular aging. Gender differences in stiffness are studied in some papers in the context of age-related changes in sex hormones, diet, AH, and exercise [19,20]. However, studies in passive orthostatic tests standardized by hydrostatic column height have not yet been conducted.

The aim of the present study is to investigate gender differences in orthostatic increase in vascular stiffness during the head-up tilt test (HUTT), creating a standard hydrostatic load.

## 2. Material and Methods

### 2.1. Participants

The study was conducted in the Laboratory of Functional Haemodynamics of Smolensk State Medical University during the academic year 2023–2024 in a room with minimal noise and an ambient temperature of 24 °C to 25 °C. The study population consisted of 133 adults aged 18–20 years (93 females, 40 males) undergoing an annual preventive medical examination (Table 1).

Inclusion criteria: young adults with clinical and laboratory parameters within the normal range for their age, non-smokers, and those with no coffee or alcohol consumption 24 h before the study. Female participants were examined in the mid-follicular phase of their menstrual cycle. Individuals with any acute or chronic disease, cardiac arrhythmia, orthostatic intolerance, peripheral oedema, evidence of thrombophlebitis or varicose veins, or a body mass index greater than 30 kg/m^2^ were excluded from the study.

Limitation: The methodological flaws of the study include the uneven distribution of groups by gender, the lack of randomization, and the relatively small sample size. The observation group consisted of students whose physical activity levels were standard and did not include those engaged in sports, so we did not assess physical activity levels. Individuals with a history of blood pressure values above 140/90 mmHg were not included in the study.

### 2.2. Head-Up Tilt Test (HUTT) Procedure

HUTT was performed using an electrically powered tilting table. Measurements were taken after a 5 min interval in the horizontal position, after a 5 min interval in the tilted position, and 5 min after returning to the baseline position. The individual tilt angle was determined to establish the standardized hydrostatic column height of all participants according to the Luanda HUTT protocol [15].

### 2.3. Hemodynamic and Vascular Stiffness Measurements

All measurements at the brachial and ankle arteries, as well as the calculation of pulse wave velocity in the areas between the brachial and ankle arteries (baPWV), were conducted using a multichannel sphygmomanometer ABI-System 100 PWV (BOSO, Berlin, Germany). The accuracy of the calculations of this device was confirmed [21]. The initial measurements were performed three times in the initial horizontal position; the result of the first measurement was disregarded, and the results of the two subsequent measurements were averaged. The side with the highest systolic blood pressure (SBP) at the brachial artery was selected as the baseline for calculations. During the HUTT, blood pressure (BP), heart rate (HR), and baPWV measurements were performed twice on the reference side, and the data from the two measurements were averaged. Additional calculation of vascular stiffness indices—functional reserve of orthostatic regulation (FR) of blood circulation (difference of baPWVt during HUT and at baseline baPWVb = baPWV) and functional reserve coefficient (FRC) = baPWVt − baPWVb/baPWVb was performed.

### 2.4. Statistical Analysis

Statistical computations were performed using the STATISTICA 10 program (TIBCO Software, Palo Alto, CA, USA). Absolute values of the data were used for description. When data were normally distributed, they were combined into a variation series. Medians and quartiles (25–75% of the interquartile range) were used for quantitative data that did not have a normal distribution, determined by the Kolmogorov–Smirnov test, and the Mann–Whitney U-test was used to compare two independent groups. Pearson’s correlation was used to analyze the relationship between variables. Correlation analysis was conducted to assess the degree of association between height, weight, and body mass index (BMI) and baPWV in the baseline position, Functional Reserve Coefficient—FRC, and ΔbaPWV. The significance of the correlations was evaluated at the *p* < 0.05 level.

## 3. Results

A total of 133 young adults aged 18–20 years were examined and divided into two gender groups female and male. Descriptive and comparative characteristics of the subjects are presented in Table 1.

Young adults of different gender groups included in the study did not differ in age. Men had significantly higher height, weight, and BMI (Table 1). All baseline parameters in both gender groups were within the normal range for age-related values. However, only HR, DBP, and baPWV were identical in both groups. The remaining indices were significantly higher in males but remained within the normal range. In contrast to the baseline values, the orthostatic changes in many indices were significantly higher in men (Table 2).

During the HUTT procedure, at a standard hydrostatic load of 130 cm, both groups demonstrated a significant increase in HR. However, this increase was more pronounced in the female cohort (*p* = 0.002676). Concurrently, there was a notable decline in SBP levels across both groups, yet the values remained elevated in males (*p* = 0.000001). The baseline DBPb was identical in both groups. During the HUTT, there was a decrease in both groups, but the levels remained significantly higher in men (*p* = 0.012426). After returning to baseline, all indices returned or were close to baseline values (Table 2).

In alignment with the primary objective of our investigation, we conducted an analysis of orthostatic alterations in baPWV across distinct gender groups. No statistically significant differences were observed in the baseline values recorded in the horizontal position (*p* = 0.379807). These values were consistent with the typical age-related values observed in the general population [17]. During HUTT under standard hydrostatic loading, there was a significant increase in this index from 8.5 [7.8;9.0] to 13.4 [12.4;14.2] m/s in women and a greater increase from 8.4 [8.0;9.0] to 14.2 [13.3;15.2] m/s in men (*p* = 0.00073). The discrepancy in performance between the HUTT and the horizontal position (ΔbaPWV) has significant prognostic implications for both women and men. In women, the value of ΔbaPWV was 4.9 [4.2;5.5], while in men it was significantly higher (5.9 [4.9;6.3], *p* = 0.001). The functional reserve coefficient (FRC) is a more accurate measure of age-related changes in arterial stiffening, with higher values observed in men (FRC = 0.65 [0.57;0.75] vs. 0.57 [0.5;0.67] in women, *p* = 0.006). Upon returning to the horizontal position, baPWV returned to its baseline value in both groups.

Given the significant difference in height between the gender groups (F = 164 [160;169] vs. M = 180 [176;185], *p* = 0.000), a special statistical analysis was conducted to investigate the impact of height on the orthostatic increase in vascular stiffness in both groups. For this purpose, the female cohort was divided into two categories based on height: up to 165 cm and above 165 cm, and up to 180 cm and above 180 cm for the male cohort (Table 3 and Table 4).

Statistical analysis of the orthostatic increase in arterial stiffness in young men and women of different height showed no significant differences at baseline or during HUTT (see Table 3 and Table 4). This result allows a reliable assessment of the gender differences presented in Table 2.

Additionally, correlational analysis was conducted to determine the presence of a relationship between FRC, ΔbaPWV, and baPWVb on the one hand, and height, weight, and BMI on the other, both in the combined group and separately for men and women.

The findings revealed a weak positive correlation between height and FRC in the combined group (r = 0.283, *p* < 0.05) and in the female group (r = 0.218, *p* < 0.05). Additionally, a moderate positive correlation was observed between height and ΔbaPWV in the combined group (r = 0.322, *p* < 0.05) and a weak positive correlation between height and ΔbaPWV in the female group (r = 0.233, *p* < 0.05). The parameters of weight and BMI did not show statistically significant correlations.

Concurrently, a robust positive correlation was observed between weight and BMI, and a moderate positive correlation between weight and height, in both the combined group and the gender groups. Furthermore, a significant and reliable difference was identified between the heights of males (180 [176;185] cm) and females (164 [160;169] cm) (Table 1). Consequently, it can be reasonably concluded that in the studied gender groups, the differences in FRC, ΔbaPWV, and baPWVb are due to gender-related physiological differences rather than to height, weight, and BMI.

## 4. Discussion

Vascular ageing is an evolutionary process. This process has been the subject of extensive study in populations up to 30 years of age and older [22,23,24]. The evolution of this process is observed to occur in several distinct types, namely normal, supernormal, and early (accelerated) [25,26,27]. The diagnosis of the ageing phenotype has significant theoretical and practical implications, particularly considering the growing prevalence of age-related vascular diseases. It is crucial to highlight that the process of age-related evolution is markedly accelerated by the onset of various diseases, including diabetes mellitus, arterial hypertension, kidney disease, chronic inflammatory conditions, and numerous others [28,29,30,31]. This underscores the significance of early diagnosis of vascular stiffness indices in a young, healthy population, prior to age-related structural changes in the vascular wall and the development of comorbid conditions that accelerate their progression.

Vascular stiffness, as measured by carotid–femoral PWV and baPWV across different gender and age groups, is well-documented [17,32]. Studies show that vascular stiffness is slightly higher in women before the age of 50, but after this threshold, the index becomes more pronounced in men [17]. The distinctive aspect of our study is its focus on not only the gender disparities in arterial stiffening but also the evaluation of the characteristics of orthostatic regulation of blood circulation. To this end, we employed a protocol (HUTT) that involved standardized hydrostatic loading. As previously stated, clinical and preclinical orthostatic disorders have been demonstrated to elevate the risk of age-related vascular disease. This information was obtained through the utilization of the active standing test. In the context of normal circulatory regulation, significant individual differences in height are irrelevant. In clinical or preclinical orthostatic hypotension (OH) and orthostatic hypertension (OHT) with impaired orthostatic regulation, these factors are relevant because they are associated with early remodeling and increased vascular stiffness compared with ONT [13,16]. Standardizations of hydrostatic loading removes this limitation and provides a more accurate assessment of the results of orthostatic regulation of blood circulation.

Another novel aspect of our study is the use of dynamic biomarkers of vascular reserve, such as ΔbaPWV and functional reserve coefficient (FRC) [18]. These indices represent external manifestations of internal adaptive processes observed in the cardiovascular system when hydrostatic pressure increases. Given their high prognostic significance, it is essential to provide a more detailed account of the physiological processes that directly influence their values.

A change in the position of the human body from horizontal to vertical results in the redistribution of blood, caused by the summation of systemic and hydrostatic pressure in the lower parts of the vascular system. The result of this summation is a significant increase in blood pressure in the distal leg veins, reaching six times the initial value, and in distal leg pressure, which rises by 70% [33]. This significant increase in intravascular pressure in a healthy person leads to the deposition of some blood in the lower parts of the vascular system. This can result in an excessive accumulation of blood in the lower extremities, typically reaching approximately 800 mL [34]. The veins and arteries of the lower limbs limit excessive blood deposition by increasing the vascular smooth muscle tone. The significance of such tone increase in veins is significantly reduced due to a small smooth muscle layer; however, veins are surrounded by muscles involved in maintaining the human body in an upright position. Furthermore, venous valves and external pressure (muscle pump) facilitate venous return to the heart [35,36]. Meanwhile, arteries in the lower limbs compensate through increased smooth muscle tone. These important adaptive processes highlight the need to understand the physiological responses to blood redistribution in orthostasis.

When hydrostatic pressure rises (e.g., in orthostasis), blood volume redistributes to the lower vascular system, reducing venous return to the heart. This insufficient filling of the right atrium triggers sympathetic baroreceptors located in the right atrium, ventricles, and pulmonary arteries [37,38,39,40]. This results in the activation of the sympathetic part of the autonomic nervous system, the sympathoadrenal system, as well as the renin–angiotensin–aldosterone system and the hypothalamic-pituitary system [41,42,43,44,45,46]. A direct consequence of the increased activity of the pressor systems is an increase in arterial smooth muscle tone and peripheral resistance, which prevents a fall in blood pressure that could lead to cerebral hypoperfusion and syncope.

The orthostatic increase in arterial smooth muscle tone is accompanied by an increase in arterial stiffening, which we assess using baPWV. The higher this index is, the greater is the ability of the vascular system to adapt to orthostatic redistribution of blood, and vice versa, a decrease in this index is expected to result in a decrease in the adaptive capacity of this system. Therefore, this index was defined as the functional reserve of the orthostatic regulation of blood circulation. Our hypothesis was indirectly confirmed in a recent study of healthy adults in three age groups [16]. The study showed not only an age-related increase in vascular stiffness but, more importantly, a decrease in functional reserve with age. These data led to the development of a new indicator of ‘vascular ageing’, which considers both the increase in vascular stiffness due to structural changes in the vascular wall and the age-related decrease in functional reserve. However, it remains unclear whether impaired orthostatic regulation, manifested by a decrease in dynamic biomarkers of vascular reserve (ΔbaPWV and FRC) is primary or whether these changes are due to an age-related increase in baseline vascular stiffness (baPWVb). Prospective cohort studies with large samples of healthy subjects are needed to definitively answer these questions.

The present study showed gender differences in ΔbaPWV and FRC, which were significantly higher in young men than in young women. The difference in these values between the sexes is probably explained by the greater training of the cardiovascular system in men due to higher physical activity. Baseline stiffness values according to baPWV data were almost identical. This may reflect the potentially high power of dynamic biomarkers. Thus, we obtained evidence of high informativeness of the dynamic marker baPWV in a young healthy population.

The functional reserve coefficient becomes even more valuable in older age groups. Furthermore, this index may be important for detecting early signs of accelerated vascular aging as reciprocal age-related dynamics of functional reserve ΔbaPWV and initial baPWVb make it possible to detect this process even with a minimal increase in this index within normal age-related values. This indicator will be useful for assessing the effectiveness of preventive measures, including those related to nutrition, physical activity, avoidance of harmful habits, etc. It also facilitates the evaluation of therapies aimed at slowing down vascular ageing. In patients who have reached the clinical stage of the disease, it is possible to assess the effectiveness of treatment in terms of reducing vascular stiffness and increasing the functional reserve of orthostatic regulation of blood circulation.

The continuing high prevalence of age-related vascular disease, and the significant mortality, disability, and economic loss associated with it, underlines the importance of finding new solutions in primary prevention. The focus of such preventive measures should be on young, healthy individuals. This requires the use of a tailored methodological approach to identify predictors and risk factors associated with future disease, not only from a mathematical perspective but also in terms of their underlying pathogenesis. The peculiarities of orthostatic regulation of blood circulation in both young and elderly populations represent risk factors for age-related vascular disease and are associated with an early increase in vascular stiffness. In the present study, we determined informative parameters—dynamic biomarkers, namely functional reserve (ΔbaPWV) and functional reserve coefficient (FRC), which allow the identification of gender-specific features of orthostatic increase in vascular stiffness in healthy young people. The results indicate the potential value of these parameters in the diagnosis of accelerated ageing and the evaluation of the efficacy of preventive measures at the preclinical stage. The incorporation of personalized monitoring of dynamic biomarkers into the system of annual preventive examinations may serve as a crucial element of a novel, effective primary prevention strategy for age-related vascular disease.

## 5. Conclusions

In our study in young adults, no significant differences in baPWVb values were found in different gender groups, but significantly higher values of dynamic biomarkers of vascular reserve were observed in men. The high reliability of intergroup differences in ΔbaPWV and FRC in our study suggests their high informativeness in personalized monitoring of vascular ageing in a young population and for objectification of the dynamics of the individual state of the vascular system against the background of lifestyle modification, preventive measures, and treatment. It is necessary to conduct a prospective cohort study to investigate the age progression of dynamic biomarkers of the functional reserve of orthostatic regulation of hemodynamics in different gender groups, their association with different phenotypes of orthostatic circulatory disorders, and with the risk of developing AH and other age-related CVDs.

At clinical stages of diseases, the proposed method will allow us to assess the effect of treatment on slowing down the rate of vascular aging and on increasing or stabilizing the functional reserve of orthostatic regulation of circulation.

## Figures and Tables

**Table 1 diagnostics-15-00517-t001:** Characterization of gender groups of subjects.

	Age Years	Height cm	Weight kg	Body Mass Index
Female (*n* = 93)	18 [18;19]	164 [160;169]	57 [52;65]	21.1 [19.1;22.6]
Male (*n* = 40)	18 [18;20]	180 [176;185]	74.5 [64;82]	22.5 [20.4;25.2]
*p*	0.556	0.000	0.000	0.025

Note: The data are presented as Me [range]—median—for parameters that were not normally distributed, quartiles—[25%; 75%]—margins of the interquartile range, *p*—the significance of inter-group differences in selected characteristics.

**Table 2 diagnostics-15-00517-t002:** Gender differences in orthostatic changes of BP, HR, and baPWV (Me [25%; 75%] indices during HUTT standardized by hydrostatic column height.

Parameters	Female	Male	*p*
	*n* = 93	*n* = 40	
HRb b/min	71 [64;78]	67 [60;75]	0.157
HRt b/min	82 [75;89]	75 [69;84]	0.003
HRr b/min	66 [60;71]	65 [59;71]	0.456
SBPb mmHg	113 [108;117]	122 [115;129]	0.000
SBPt mmHg	104 [99;110]	114 [108;125]	0.000
SBPr mmHg	112 [106;117]	122 [115;129]	0.000
DBPb mmHg	70 [65;74]	69 [65;73]	0.446
DBPt mmHg	68 [63;72]	71 [67;78]	0.012
DBPr mmHg	70 [64;74]	71 [65;74]	0.546
baPWVb, m/s	8.5 [7.8;9.0]	8.4 [8.0;9.0]	0.380
baPWVt, m/s	13.4 [12.4;14.2]	14.2 [13.3;15.2]	0.001
baPWVr m/s	8.4 [7.8;9.1]	8.7 [8.2;9.3]	0.507
ΔbaPWV, m/s	4.9 [4.2;5.5]	5.9 [4.9;6.3]	0.001
FRC	0.57 [0.5;0.67]	0.65 [0.57;0.75]	0.006

Note: SBP—systolic blood pressure brachial; DBP—diastolic blood pressure brachial; HR—heart rate; baPWV—the brachial–ankle pulse wave velocity. The values of these indicators in the initial position are marked with ‘b’—baseline; in tilt up position—‘t’, after returning to the baseline position ‘r’—recovery; the significance of intergroup differences is *p* < 0.05. Functional reserve ΔbaPWV = baPWVt − baPWVb, Functional Reserve Coefficient FRC = ΔbaPWV/baPWVb.

**Table 3 diagnostics-15-00517-t003:** Particularities of orthostatic changes in baPWV in young women of different heights.

Parameters	Height up to 165 cm	Height over 165 cm	*p*
	*n* = 50	*n* = 43	
baPWVb m/s	8.5 [7.9;9.0]	8.5 [7.7;9.0]	0.997
ΔbaPWV m/s	4.7 [4.0;5.4]	5.0 [4.4;5.6]	0.300
FRC	0.57 [0.5;0.7]	0.6 [0.5;0.7]	0.361

Note: The data are presented as Me [range]—median—for parameters that were not normally distributed, quartiles [25%; 75%]—margins of the interquartile range, baPWV—the brachial–ankle pulse wave velocity. Functional reserve ΔbaPWV = baPWVt − baPWVb, Functional Reserve Coefficient FRC = ΔbaPWV/baPWVb.

**Table 4 diagnostics-15-00517-t004:** Particularities of orthostatic changes in baPWV young men of different heights.

Parameters	Height up to 180 cm	Height over 180 cm	*p*
	*n* = 19	*n* = 21	
baPWVb m/s	8.7 [8.3;9.4]	8.2 [7.9;9.0]	0.072
ΔbaPWV m/s	5.95 [4.6;6.7]	5.8 [5.0;6.1]	0.597
FRC	0.63 [0.58;0.75]	0.67 [0.58;0.76]	0.871

Note: The data are presented as Me [range]—median—for parameters that were not normally distributed, quartiles [25%; 75%]—margins of the interquartile range, baPWV—the brachial–ankle pulse wave velocity. Functional reserve ΔbaPWV = baPWVt − baPWVb, Functional reserve Coefficient FRC—ΔbaPWV/baPWVb.

## Data Availability

Data are contained within the article.

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
