# Peer review of "Gender Difference in Orthostatic Vascular Stiffness Increase in Young Subjects"

_diagnostics, 2025, doi:10.3390/diagnostics15050517_

Round 1

Reviewer 1 Report

Comments and Suggestions for Authors

The study investigates gender differences in orthostatic vascular stiffness and aging among healthy young adults 

1. In general, the use of [;] to denote intervals is unconventional and may lead to confusion. A more standard format, such as (min–max) or [min–max], should be used for clarity and consistency with common practices. Also, the p-value is usually standardized to three decimal places for consistency.

2. Table 1 : The table's horizontal and vertical arrangement needs to be swapped, and the titles should be clarified. Additionally, abbreviations like 'Me' should not appear as titles but should be written out in full below the table.

3. Line 147

There are empty parentheses () within the text, which may indicate missing information or errors. These should be reviewed and corrected to ensure clarity and completeness

Additional comments:

There is no adjustment for baseline characteristics, which raises concerns about the validity of the conclusions drawn. Differences in baseline variables, such as height, weight, or BMI, may significantly influence the results and should be accounted for to ensure robust and reliable findings.

Author Response

Thank you very much for taking the time to review this manuscript/ Please find the detailed responses and the corresponding revisions 

Comment 1: In general, the use of [;] to denote intervals is unconventional and may lead to confusion. A more standard format, such as (min–max) or [min–max], should be used for clarity and consistency with common practices. Also, the p-value is usually standardized to three decimal places for consistency.
Response 1: In our study, non-parametric data were used, and for their visualization, median values and quartiles [Q1; Q3] or [25%; 75%] were employed, as presented in the work. This methodology does not involve indicating minimum and maximum values, as quartiles provide a more informative representation of the data distribution. Moreover, the use of [;] to denote intervals is considered standard in scientific publications, especially when presenting quartiles and other statistical measures. This format allows for the precise conveyance of information about data distribution and ensures consistency with accepted standards in the field.
We standardized the p-value in tables and in text to three decimal places.

Comment 2 Table 1 : The table's horizontal and vertical arrangement needs to be swapped, and the titles should be clarified. Additionally, abbreviations like 'Me' should not appear as titles but should be written out in full below the table.

Response 2 All tables in this paper, including Table 1, are structured stereotypically with our previous publications in this series, which are referenced in this paper.
The table titles and notes have been modified according to your recommendations.

Comment 3 Line 147
There are empty parentheses () within the text, which may indicate missing information or errors. These should be reviewed and corrected to ensure clarity and completeness/

Response 3 Thanks for the comment, that typo has been corrected (line 159).

Additional comments:

There is no adjustment for baseline characteristics, which raises concerns about the validity of the conclusions drawn. Differences in baseline variables, such as height, weight, or BMI, may significantly influence the results and should be accounted for to ensure robust and reliable findings.

Response: The present study sought to analyse the correlation between FRC, ΔbaPWV, and baPWVb on the one hand, and height, weight, and body mass index (BMI) on the other, both in the combined group and separately for men and women. 
The findings revealed a weak positive correlation between height and FRC in the combined group (r = 0.283, p < 0.05) and in the female group (r = 0.218, p < 0.05). Additionally, a moderate positive correlation was observed between height and ΔbaPWV in the combined group (r = 0.322, p < 0.05) and a weak positive correlation between height and ΔbaPWV in the female group (r = 0.233, p < 0.05).No significant correlations of FRC, ΔbaPWV, baPWVb with weight and body mass index (BMI) values were found.
Concurrently, a robust positive correlation was observed between weight and BMI, and a moderate positive correlation between weight and height, in both the combined group and the gender groups. Furthermore, a significant and reliable difference was identified between the heights of males (180[176;185] cm) and females (164[160;169] cm). Consequently, it can be reasonably concluded that in the studied gender groups, the differences in FRC, ΔbaPWV, and baPWVb are due to gender-related physiological differences rather than to height, weight, and BMI.

Reviewer 2 Report

Comments and Suggestions for Authors

In this study of young adults, baseline brachial-ankle pulse wave velocity (baPWVb) values did not significantly differ between genders. However, men exhibited significantly higher values for dynamic biomarkers of vascular reserve, including ΔbaPWV and FRC. These findings highlight the potential of these biomarkers for personalized monitoring of vascular aging, especially in younger populations, where early detection is crucial.

However, some concerns have been raised.

1.     These values were consistent with the typical age-related values observed in the general population (). Please correct it in ().

2.     What are the reasons or references for checking the height up to 165 cm and above 165 cm for the female and up to 180 cm and above 180 cm for the male?

3.     The analysis mentions gender differences but does not explore how differences in physical activity levels, hormonal influences, or height might confound the results. Addressing these factors would provide a more comprehensive discussion.

4.     While the statistical methods are described, there is no mention of how potential outliers were handled or if there was an adjustment for multiple comparisons. Including this information would improve the methodological rigor.

5.     The ΔbaPWV and functional reserve coefficient (FRC) did not prove to be associated with cardiovascular risk or mortality or other risk factors of diabetes, stroke, hypertension, or MACE. How are these parameters for clinicians for clinical use?  How to establish their normative ranges, predictive value, and broader applicability across diverse populations.

6.     The limitations of this study and discuss the feasibility of applying the findings in broader populations or different age groups.

7.     baPWV is highly sensitive to blood pressure at the time of measurement. Elevated blood pressure can artificially increase baPWV, making it difficult to distinguish between acute hemodynamic changes and chronic structural stiffness. Height and body composition, age, and sex also affected the baPWV level. Please describe why not adjust these cofounders in this study.

Author Response

Thank you very much for taking the time to review this manuscript. Please find the detailed responses below and the corresponding revisions.

Comment 1 These values were consistent with the typical age-related values observed in the general population (). Please correct it in ().
Response 1 Literature reference [17] inserted (line 159).
Comment 2.    What are the reasons or references for checking the height up to 165 cm and above 165 cm for the female and up to 180 cm and above 180 cm for the male?
Response 2. There were significant differences in height between the male and female groups (see Table 1). This suggests the question of whether the sex differences in stiffness could be related to the significant height differences. To answer this, we divided each group into low and high: in women, these were subgroups up to and including 164 cm and 165 cm and above, and in men, up to and including 179 cm and 180 cm and above. The fact that there were no differences in the studied indicators between subgroups of different height suggests that the differences between the groups are not due to the difference in height between men and women. The results of the study showed no link between height and the vascular stiffness measures that were looked at. This was true both when the subjects were at rest and during the HUTT.
Comment 3. The analysis mentions gender differences but does not explore how differences in physical activity levels, hormonal influences, or height might confound the results. Addressing these factors would provide a more comprehensive discussion.
Response 3. In our study, we did not set the task of studying how athletes' training affects their blood pressure when they are standing up. We examined medical university students who had undergone a complete check-up, had no physical activity limitations, and did not have obesity or cachexia. The lifestyle of all subjects was approximately similar. We examined the possible influence of height (see above answer to question 2).The present study sought to analyse the association of FRC, ΔbaPWV, and baPWVb with height, weight, and body mass index (BMI) values, both in the combined group and separately for men and women. Furthermore, the association of FRC, ΔbaPWV, and baPWVb with subject height was analyzed, both in the combined group and separately for men and women.The findings revealed a weak positive correlation between height and FRC in the combined group (r = 0.283, p < 0.05) and in the female group (r = 0.218, p < 0.05). Additionally, a moderate positive correlation was observed between height and ΔbaPWV in the combined group (r = 0.322, p < 0.05) and a weak positive correlation between height and ΔbaPWV in the female group (r = 0.233, p < 0.05). In addition, a significant and reliable difference was observed between the heights of males (176–185 cm) and females (160–169 cm). Therefore, it can be reasonably concluded that, in the gender groups studied, the differences in FRC, ΔbaPWV, and baPWVb are due to gender physiological differences rather than to the height of the subjects.
Comment 4. While the statistical methods are described, there is no mention of how potential outliers were handled or if there was an adjustment for multiple comparisons. Including this information would improve the methodological rigor.
Response 4. In our study, comparisons were made based on gender and growth indicators. These data were presented in tables for ease of understanding. Since there were only two values in each case, we did not need to use adjustments for multiple comparisons. Regarding outliers, we applied statistical methods such as the Mann-Whitney U-test, which is used to compare two independent groups and considers all values in the studied groups during the calculation.
Comment 5. The ΔbaPWV and functional reserve coefficient (FRC) did not prove to be associated with cardiovascular risk or mortality or other risk factors of diabetes, stroke, hypertension, or MACE. How are these parameters for clinicians for clinical use?  How to establish their normative ranges, predictive value, and broader applicability across diverse populations.
Response 5. You are right, our study was performed on healthy subjects under 20 years of age. ΔbaPWV and FRC are not directly related to cardiovascular risk, and this is very important!Risk and vascular aging (increased arterial stiffening) is assessed in the horizontal position, at rest. This is shown by changes in vascular wall structure, increased pulse wave velocity, as well as intima-media thickness and other parameters. Our study looks at controlling vascular ageing before any changes to the arteries happen, when baPWV and other measurements are normal for age. The ΔbaPWV and FRC measurements show how stiff the arteries are. They show how well the body can handle changes in pressure when you change position. These measurements are a sign of how the body controls blood flow when you change position. Problems with this regulation, like in cases of orthostatic hypotension and hypertension, can be a risk factor for AH and are linked to early remodeling and increased stiffness of the vascular wall [13,16]. The new thing about our work is that we have suggested indicators that allow: We can diagnose early disorders of orthostatic regulation in young, healthy subjects. In this part, we can clarify the normal ranges of these indicators for different age groups and how useful they are for predicting outcomes. This will be done through a representative prospective cohort study, which is already being implemented. 2) We can objectify the individual dynamics of the cardiovascular system's ability to adapt to a standard change in hydrostatic load. This includes how it responds to certain preventive or therapeutic measures, or the absence of the latter. The indicators we suggest can help patients choose the best treatment for them. In the long term, they can show how well the treatment works for everyone.
Comment 6. The limitations of this study and discuss the feasibility of applying the findings in broader populations or different age groups.
Response 6 It is accurate to observe that the present study is limited in scope, a limitation that is directly related to the objectives of the study. The investigation revealed gender disparities in the orthostatic regulation of blood circulation in young, healthy subjects, and the study demonstrated the informative nature of novel functional markers. Consequently, a baseline has been established, representing a preliminary starting point for the study of the age progression of vascular ageing. This is a prerequisite for the identification of healthy subjects who exhibit early and accelerated vascular ageing. It is precisely these subjects who will require greater attention in primordial prophylaxis.
Comment 7. baPWV is highly sensitive to blood pressure at the time of measurement. Elevated blood pressure can artificially increase baPWV, making it difficult to distinguish between acute hemodynamic changes and chronic structural stiffness. Height and body composition, age, and sex also affected the baPWV level. Please describe why not adjust these cofounders in this study. 
Response 7. The investigation revealed gender disparities in the orthostatic regulation of blood circulation in young, healthy subjects, and the study demonstrated the informative nature of novel functional markers. Consequently, a baseline has been established, representing a preliminary starting point for the study of the age progression of vascular ageing. This is a prerequisite for the identification of healthy subjects who exhibit early and accelerated vascular ageing. It is precisely these subjects who will require greater attention in primordial prophylaxis. We conducted a correlation analysis between FRC, ΔbaPWV, and baPWVb on the one hand, and height, weight, and body mass index (BMI) on the other, both in the combined group and separately for men and women. The findings revealed a weak positive correlation between height and FRC in the combined group (r = 0.283, p < 0.05) and in the female group (r = 0.218, p < 0.05). Additionally, a moderate positive correlation was observed between height and ΔbaPWV in the combined group (r = 0.322, p < 0.05) and a weak positive correlation between height and ΔbaPWV in the female group (r = 0.233, p < 0.05).No significant correlations of FRC, ΔbaPWV, baPWVb with weight and body mass index (BMI) values were found. Concurrently, a robust positive correlation was observed between weight and BMI, and a moderate positive correlation between weight and height, in both the combined group and the gender groups. Furthermore, a significant and reliable difference was identified between the heights of males (180[176;185] cm) and females (164[160;169] cm). Consequently, it can be reasonably concluded that in the studied gender groups, the differences in FRC, ΔbaPWV, and baPWVb are due to gender-related physiological differences rather than to height, weight, and BMI.

Reviewer 3 Report

Comments and Suggestions for Authors

The presented study compared orthostatic increases in vascular stiffness during the head-up tilt test, in young healthy men and women. For this purpose, the authors prospectively examined and included in the study 133 healthy adults aged 18–20 years, among which there were 90 women and 40 men. To analyze the data, the authors introduced a new coefficient - the functional reserve coefficient (FRC) of orthostatic circulatory regulation. In addition, the authors investigated particularities of orthostatic changes in baPWV depending on height. According to research results in young adults, no significant differences in baPWVb values were found in different gender groups, but significantly higher values of FRC were observed in men. The presented study is interesting in that the authors proposed a new, easy-to-assess coefficient, which allows performing personalized monitoring of vascular ageing. The article is well-structured and clearly written. The main limitation of the study is the small sample size, and very heterogeneous groups, varying significantly in the number of included persons (men are twice less than women) and their basic characteristics (especially baseline systolic pressure). In this regard, I recommend to apply a propensity score-based adjustment of the groups and to repeat an intergroup comparison in the propensity-matched population.

Minor comments.

1.      The title does not fully correspond to the content of the article, since the study was performed on a young population and vascular aging was not directly studied.

2.      The introduction of both the abstract and the text of the article does not provide sufficient background, since there is no literature references, concerning gender differences in vascular stiffness and the mechanisms of their formation (e.g. 10.1007/s10286-022-00911-z., 10.1111/bph.14624,  10.1186/s13293-020-00294-8)

3.      According to the data in Table 2, some of the men had elevated baseline SBP (more than 120 mm Hg). Could this have affected the results of the study?

4.      What was the purpose of tilt test in healthy patients? This method is not routinely used in medical preventive examinations.

5.      There is no information about FRC calculation in the Materials and methods (only in the Introduction). It should be added.

How the point for dividing patient groups into subgroups by height was defined? I think it is more correct to evaluate correlations between height and vascular stiffness indicators.

Author Response

Thank you very much for taking the time to review this manuscript. Please find the detailed responses below and the corresponding revisions.
Comment 1. The title does not fully correspond to the content of the article, since the study was performed on a young population and vascular aging was not directly studied.
Response 1. It is acknowledged that the investigation of vascular aging in young, healthy subjects was not a primary focus of this study. The central objective was to examine gender disparities in orthostatic increase in vascular stiffness. The investigation encompassed healthy subjects under 20 years of age, with a particular focus on ΔbaPWV and FRC, domains in which prior studies have revealed significant inter-age group variations [18]. The conceptual framework underpinning this study is the conception of aging as a decline in the adaptive capacity of the organism. The assessment of vascular aging is conventionally undertaken by measuring resting baPWV, a metric that predominantly reflects alterations in vascular wall structure. The present study is centered on the investigation of vascular aging in its earliest stages, preceding the onset of structural changes in arteries, when baPWV and other indices are within the age-related norms.The ΔbaPWV and FRC indices studied by us are functional components of stiffness increase, which reflect orthostatic regulation of blood circulation (adaptation of the vascular system to changes in hydrostatic pressure during changes in body position).Disturbances of such regulation, as occurs in orthostatic hypotension and hypertension, are risk factors for AH and are associated with early remodeling and increased stiffness of the vascular wall, i.e. with generally recognized signs of vascular ageing. However, should the respected reviewer propose an alternative title that is deemed more appropriate, we will duly consider it.
Comment 2. The introduction of both the abstract and the text of the article does not provide sufficient background, since there is no literature references, concerning gender differences in vascular stiffness and the mechanisms of their formation (e.g. 10.1007/s10286-022-00911-z., 10.1111/bph.14624,  10.1186/s13293-020-00294-8).
Response 2. We would like to express our gratitude to the esteemed reviewer for his constructive feedback. The introduction has been augmented with information regarding gender disparities in rigidity within the context of age-related fluctuations in sex hormones, diet, AG, and physical activity. Reference [19,20] added (line 71).
Comment 3. According to the data in Table 2, some of the men had elevated baseline SBP (more than 120 mm Hg). Could this have affected the results of the study?
Response 3. Such minor fluctuations in blood pressure may exert a negligible effect on baseline baPWVb values in accordance with the Baylis effect. Notable orthostatic increases in this index are predominantly attributable to a neurohormonal orthostatic shift.
Comment 4. What was the purpose of tilt test in healthy patients? This method is not routinely used in medical preventive examinations.
Response 4. The objective of the present study was twofold: firstly, to examine gender differences in orthostatic stiffness increase, and secondly, to analyse the sensitivity of key indices of functional reserve in orthostatic regulation of blood circulation in a group of young, healthy adults. It is acknowledged that this methodology has not yet been incorporated into medical examinations. However, the necessity to develop tools to objectivize the dynamics of adaptation capabilities of the organism (in our case, the vascular system) is highly relevant. Our preliminary studies have revealed distinctive characteristics of vascular ageing, including both the well-documented progressive increase in vascular stiffness and the reciprocal decrease in ΔbaPWV and FRC [18]. Consequently, the study of these indices will facilitate the objective assessment of the natural dynamics of adaptation capabilities of the cardiovascular system in a particular individual. In the long term, this will also allow the evaluation of the effectiveness of certain preventive or therapeutic measures for the selection of optimal tactics.
Comment 5. There is no information about FRC calculation in the Materials and methods (only in the Introduction). It should be added. The main limitation of the study is the small sample size, and very heterogeneous groups, varying significantly in the number of included persons (men are twice less than women) and their basic characteristics (especially baseline systolic pressure). In this regard, I recommend to apply a propensity score-based adjustment of the groups and to repeat an intergroup comparison in the propensity-matched population.
Response 5. We extend our gratitude to the esteemed reviewer for their insightful commentary. The Material and Methods section has been updated to include details on the calculation of ΔbaPWV and FRC (lines 105-108).

When forming the groups, we ensured that the number of observations was at least 35 in each group and used non-parametric research methods to ensure correct data interpretation. Indeed, the number of observations in the male and female groups differs by a factor of two, and for natural reasons, they significantly differ from each other in weight and height factors, although they do not differ in age. This was a fundamental aspect of our work. Additionally, it was important in our study to demonstrate the absence of statistically significant differences in the baPWVb parameter in groups divided by gender, as well as the presence of differences in the ΔbaPWV parameter. In addition to these tables, we conducted a correlation analysis between FRC, ΔbaPWV, and baPWVb on the one hand, and height, weight, and body mass index (BMI) on the other, both in the combined group and separately for men and women. The findings revealed a weak positive correlation between height and FRC in the combined group (r = 0.283, p < 0.05) and in the female group (r = 0.218, p < 0.05). Additionally, a moderate positive correlation was observed between height and ΔbaPWV in the combined group (r = 0.322, p < 0.05) and a weak positive correlation between height and ΔbaPWV in the female group (r = 0.233, p < 0.05).No significant correlations of FRC, ΔbaPWV, baPWVb with weight and body mass index (BMI) values were found. Concurrently, a robust positive correlation was observed between weight and BMI, and a moderate positive correlation between weight and height, in both the combined group and the gender groups. Furthermore, a significant and reliable difference was identified between the heights of males (180[176;185] cm) and females (164[160;169] cm). Consequently, it can be reasonably concluded that in the studied gender groups, the differences in FRC, ΔbaPWV, and baPWVb are due to gender-related physiological differences rather than to height, weight, and BMI.

Round 2

Reviewer 2 Report

Comments and Suggestions for Authors

Please add physical activity levels and not record blood pressure in the limitations of this study. The this paper can accept.

Author Response

We have added to our study's limitations section that we did not measure the physical activity levels of the observation groups and did not include individuals with blood pressure levels above 140/90 in their medical history. Thank you!

Reviewer 3 Report

Comments and Suggestions for Authors

The authors have responded to most of the comments, but the question of randomization and comparability of the groups remains unanswered. In this regard, I propose adding a Limitations section to the article and to indicate methodological flaws of the study.

Author Response

We have added the limitations you suggested to the relevant section of the article. Thank you!